# Myocarditis and Pericarditis Following mRNA COVID-19 Vaccination: What Do We Know So Far?

**DOI:** 10.3390/children8070607

**Published:** 2021-07-18

**Authors:** Bibhuti B. Das, William B. Moskowitz, Mary B. Taylor, April Palmer

**Affiliations:** 1Department of Pediatrics, Children’s of Mississippi Heart Center, University of Mississippi Medical Center, Jackson, MS 39216, USA; wmosckowitz@umc.edu; 2Department of Pediatrics, Division of Critical Care, University of Mississippi Medical Center, Jackson, MS 39216, USA; mbtaylor@umc.edu; 3Department of Pediatrics, Division of Infectious Disease, University of Mississippi Medical Center, Jackson, MS 39216, USA; aplamer@umc.edu

**Keywords:** COVID-19 vaccine, myocarditis, pericarditis

## Abstract

This is a cross-sectional study of 29 published cases of acute myopericarditis following COVID-19 mRNA vaccination. The most common presentation was chest pain within 1–5 days after the second dose of mRNA COVID-19 vaccination. All patients had an elevated troponin. Cardiac magnetic resonance imaging revealed late gadolinium enhancement consistent with myocarditis in 69% of cases. All patients recovered clinically rapidly within 1–3 weeks. Most patients were treated with non-steroidal anti-inflammatory drugs for symptomatic relief, and 4 received intravenous immune globulin and corticosteroids. We speculate a possible causal relationship between vaccine administration and myocarditis. The data from our analysis confirms that all myocarditis and pericarditis cases are mild and resolve within a few days to few weeks. The bottom line is that the risk of cardiac complications among children and adults due to severe acute respiratory syndrome coronavirus 2 (SARS-CoV-2) infection far exceeds the minimal and rare risks of vaccination-related transient myocardial or pericardial inflammation.

## 1. Introduction

The mRNA vaccines ((BNT162b2, Pfizer-BionTech, Pfizer, Inc.; Philadelphia, PA, USA) and Moderna (mRNA-1273, ModernaTX, Inc.; Cambridge, MA, USA)) demonstrated excellent safety and clinical efficacy profiles in clinical trials in adults in less than one year from the identification of the virus which is unprecedented in the history of vaccinology [1,2]. The mRNA vaccine used in the European Union is Comirnaty, which has the same mRNA, BNT162b2- as the Pfizer-BionTech, and is manufactured according to the same processes and procedures. The only difference is in their label. The mRNA COVID-19 vaccine Moderna is also known as Spikevax, which is used in the European Union. The Food and Drug Administration (FDA) issued emergency use authorization (EUA) on 11 December 2020, for Pfizer-BioNTech COVID-19 vaccine in persons aged > 16 years [3]. The European Medicine Agency (EMA) approved Comirnaty in people from 16 years of age on 21 December 2020. On 10 May 2021, the FDA expanded EUA to use the Pfizer mRNA COVID-19 vaccine for children 12–15 years [4]. The Center for Disease Control and Prevention (CDC) subsequently recommended on 12 May 2021, that persons 12 years and older should get a COVID-19 vaccine.

Post-vaccination myocarditis has been reported as early as 1957 after smallpox vaccination [5]. Analysis of the Vaccine Adverse Event Reporting System (VAERS) data between 2011 and 2015, where a total of 357,188 reports were reviewed and found 199 cases of myocarditis and pericarditis. Only the smallpox vaccine emerged with an expectedly strong correlation with myocarditis and pericarditis [6]. Previous findings should be interpreted with caution regarding limitations affecting the voluntary reporting system and may be underreported. Conversely, in the current era of heightened surveillance by the CDC’s vaccine safety data (VSD) working group and the Vaccine Related Biological Products Advisory Committee (VRBPAC) on immunization practices after COVID-19, and post-marketing surveillance by the vaccine producing companies in the setting of conditional marketing authorization, the reporting of probable myocarditis and pericarditis cases is significantly higher. Since April 2021, increased cases of myocarditis and pericarditis have been reported in the United States after mRNA COVID-19 vaccination, particularly in adolescents and young adults [7,8,9,10,11,12].

The American Academy of Pediatrics (AAP) and the American Heart Association (AHA) have endorsed CDC recommendations and reiterated the potential benefits of COVID-19 vaccination, which outweighs rare myocarditis or pericarditis risks and recommend the vaccination for anyone 12 years of age and older [13,14]. Very little published data on the incidence of mRNA vaccine-associated myocarditis and pericarditis exist except those reported in the media. We performed a systematic search of electronic databases (PubMed, Scopus, medRxiv and bioRxV) with a goal to publish the results in the context of expanding the vaccine target population using the terms “mRNA vaccine complications”, “heart inflammation with COVID-19 vaccine”, “impact of COVID-19 vaccine in children and young adults”, “myocarditis after COVID-19 vaccination” and “pericarditis after COVID-19 vaccination”, and “myopericarditis after COVID-19 vaccination”. We found a total of 29 cases in the literature as of 26 June 2021. This review summarizes 29 published cases of vaccine-induced myocarditis in children and adults, describes a comprehensive management plan and emphasizes that rare instances of myocarditis should not preclude anyone from taking the COVID-19 vaccine.

## 2. Incidence of Vaccine Related Myocarditis/Pericarditis

As of 26 June 2021, a total of 322 million doses of vaccine were used, and thus far, 79 children aged 16 or 17 years, and 196 young adults aged 18–24 years have been confirmed by the CDC as having myocarditis/pericarditis following mRNA COVID-19 vaccination after analyzing data in the VAERS [15]. The adjusted risk ratio for myocarditis and pericarditis events in children and young adults between 16 and 24 years of age has been determined to be 0.94 (95% confidence interval 0.59–1.52) [16,17]. Because of this, the US FDA added a warning about risk of myopericarditis and pericarditis to the fact sheet of mRNA COVID-19 vaccines. Recently, the US military reported 23 patients among 2.8 million doses of mRNA COVID-19 vaccine administered [18]. While the observed number of myocarditis cases was small, the number was higher than expected among male military members after a second vaccine dose. In a report from the Israeli Ministry of Health, one in 3000 to one in 6000 men aged 16–24 who received the mRNA COVID vaccine developed myocarditis and pericarditis [19]. Ninety percent of the cases in Israel appear to be men. Although the background rate of myocarditis in this population is high, the rate following vaccination appeared to be 5–25 times higher than the background rate. The European Medicines Agency has also recently reported that myocarditis and pericarditis can occur in very rare cases following vaccination with COVID-19 vaccines Comirnaty and Spikevax. [20]. “The Committee is therefore recommending listing myocarditis and pericarditis as new side effects in the product information for these vaccines, together with a warning to raise awareness among healthcare professionals and people taking these vaccines” [20]. The European Medicine Agency’s safety committee (PRAC) has included 145 cases of myocarditis in the European Economic Area (EEA) among people who received Comirnaty and 19 cases following the use of Spikevax. As of 31 May 2021, around 177 million doses of Comirnaty and 20 million doses of Spikevax had been given in the EEA. As of end of May 2021, the incidence of myocarditis is 1 per million for both Comirnaty and Spikevax in the EEA.

## 3. Causal Relation between mRNA Vaccine and Myocarditis/Pericarditis

As of 30 May 2021, only few cases have been reported in the EEA from the EudraVigilance database: 38 for Vaxzevria (previously COVID-19 vaccine AstraZeneca) and 0 for Janssen vaccine (Ad.26.COV2.S) (Johnson & Johnson; New Brunswick, New Jersey, USA). There is absence of myocarditis and rare cases of pericarditis after receiving the non-mRNA COVID-19 vaccines, such as Vaxsevria or Janssen but conversely, cases of myocarditis and pericarditis in young men after mRNA vaccination are rising. This raises the question of why this only occurs after the vaccines based on an mRNA platform? Several mechanisms have been hypothesized. It has been speculated from the data reported in the initial trials of mRNA vaccines in adults that mRNA vaccines might generate a very high antibody response in a small subset of young people, thus eliciting a response similar to multisystem inflammatory syndrome in children (MIS-C) associated with severe acute respiratory syndrome coronavirus 2 (SARS-CoV-2) infection. [21] However, the cases reported in the literature had no data on COVID antibody test results in all cases, which is a limitation of this analysis. Other hypothesized mechanisms include induction of anti-idiotype cross-reactive antibody-mediated cytokine expression in the myocardium and the aberrant induction of apoptosis and results in inflammation of myocardium and pericardium. Furthermore, Muthukumar et al. conducted detailed immunologic investigation in a 52-year-old man who developed myocarditis three days after receiving the second dose of Moderna mRNA COVID-19 vaccine and found that his antibody responses to 18 different SARS-CoV-2 antigens did not differ (and was lower for some antigens) from vaccinated controls who did not develop complications [22]. The mRNA vaccines can also induce a non-specific innate inflammatory response or a molecular mimicry mechanism between the viral spike protein and an unknown cardiac protein [23]. The other possibilities include the RNA in the vaccine itself, a potent immunogen, and produces bystander or adjuvant effect [24] by cytokine activation of pre-existing autoreactive immune cells as young people usually have higher seroprevalence of SARS-CoV-2 even if they are asymptomatic during the COVID-19 pandemic. Molecular Koch’s postulates are used to determine what contributes to a pathogen’s ability to cause disease. In the future, mRNA used in vaccine production can be inoculated directly into human induced pluripotent stem cells-derived myocytes [25,26] to study transcriptomic and morphological changes. It may help in understanding the mechanism of the mRNA vaccine-induced myocarditis or pericarditis. The incidents of mRNA vaccine-related myocarditis and pericarditis occur predominantly in the male gender, probably related to a higher rate of vaccination in this group compared to females, as shown from Israel [19].

## 4. Clinical Summary of Published Cases of Myocarditis and Pericarditis

The Table 1 summarizes 29 total cases (both children and adults) and published in the literature as of 26 June 2021. Out of these 29 cases, four patients (cases 25–28) were from Italy, and the rest were from the US. Of them all, 13 patients are ≤18 years. All patients were male. Except for one (case 15), all vaccine-related myocarditis and pericarditis cases were after mRNA vaccination. Three (10%) cases (cases 13, 18 and 24) were only after the first dose of vaccine, and the rest were after the second dose of the vaccine. The most common symptoms were chest pain, followed by fever, and rarely dyspnea, cough and headache. After vaccination, the onset of symptoms occurred most commonly within 3 days but ranged between 1 and 7 days. It was difficult to separate myocarditis from pericarditis from the published cases, and most notably, there was no troponin data available in seven cases. All cases where troponin and C-reactive protein (CRP) were reported had elevated levels. The most common abnormal ECG finding was ST-segment elevation in 11 (38%), followed by diffuse ST changes in 10 (35%); other changes included peaked T wave (1), junctional rhythm (1), non-specific ST changes (1), ST depression (1) and normal in (4) patients. Echocardiography showed mild dysfunction in 13 patients, of whom most were age > 18 years. There was heterogeneity in the echocardiographic findings. The patients who were ≥30 years of age had reduced left ventricle (LV) function in 66% (4/6) cases, whereas LV function was reduced in 34% (8/23) in young adults < 30 years of age. In children ≤ 18 years where an echocardiogram was available, the LV function was primarily normal in all patients. Pericardial effusion was found by echocardiogram only in two patients. Cardiac magnetic resonance imaging (CMR) was available in 20 (69%) patients, and the presence of late gadolinium enhancement (LGE) was present in all reported cases. In contrast, additional evidence of myocardial edema on T2 mapping was present only in seven (35%) patients. In most cases, patients required hospitalization for 2–7 days, a median of only 3 days. All patients recovered and were discharged home in less than a week. Two patients aged > 18 years (cases 28 and 29) had non-sustained ventricular tachycardia during hospitalization. Only one patient aged > 18 years (case 28) required inotropes and mechanical circulatory support. Most patients were treated with non-steroidal anti-inflammatory drugs (NSAID) alone, followed by NSAID with colchicine and NSAID with steroids. In patients ≤ 18 years of age where treatment was reported, four patients received intravenous immunoglobulin (IVIG) and steroids. In all cases where follow-up was available, documented complete clinical recovery in 1–3 weeks during follow-up.

## 5. Diagnosis and Management

The common clinical presentation of COVID-19 vaccine-associated myopericarditis includes chest pain, fever, palpitation, shortness of breath, fatigue, nausea, vomiting, abdominal pain or unusual symptoms such as forceful or thumping heart beats. Common signs of myopericarditis include tachypnea, tachycardia, murmur, gallop, diminished pulses, hypotension, hepatomegaly, edema and signs of low cardiac output [27]. The laboratory tests commonly used include testing to detect any viral causes of myocarditis, serum concentrations of an inflammatory marker (C-reactive protein, erythrocyte sedimentation rate) and cardiac biomarker (troponin, brain type natriuretic peptide), electrocardiogram, echocardiogram, CMR and serologic testing for SARS-CoV-2 antibodies. Historically, the diagnosis of myocarditis is confirmed by histologic criteria, including acute myocyte injury with inflammatory cells’ infiltration, especially lymphocytes [28]. A paradigm shift in the diagnosis of myocarditis has occurred [29]. According to an AHA statement, four strata of diagnosis of myocarditis in children are recommended: biopsy proven, clinically suspected, confirmed by CMR and possible myocarditis. The shift in diagnosis acknowledges advancements in CMR and improvement in identifying a constellation of clinical signs and symptoms supportive of myocarditis.

In most cases, COVID-19 vaccine-associated myocarditis is transient and self-limited; it is not justifiable to obtain an endomyocardial biopsy. Furthermore, the clinical impact of myocarditis varies widely due to the range of etiologies and the unpredictable physiologic responses depending upon the host’s response to the inciting agent. Elevated cardiac troponin may indicate acute cardiac injury, but not specific to the diagnosis of myocarditis, as many myocarditis patients are asymptomatic and are subclinical. Furthermore, features of myocarditis and pericarditis may overlap and commonly present as myopericarditis. Because of variable clinical manifestations of myocarditis, it is essential to follow the CDC definition of acute myocarditis (Figure 1) and acute pericarditis (Figure 2). Cardiac magnetic resonance imaging with tissue characterization using T1 and T2 mapping is a useful non-invasive modality for diagnosing myocarditis [30]. Localized or generalized myocardial edema on T2 recovery images without evidence of late gadolinium enhancement (LGE) and no other clinical features can be the only CMR evidence of myocardial inflammation in mRNA COVID vaccine associated myocarditis in early stages. The AHA recommended “everyone to keep in touch with their primary care professional and seek care immediately if they have any of these symptoms in the weeks after receiving the COVID-19 vaccine: chest pain including sudden, sharp, stabbing pains; difficulty breathing/shortness of breath; abnormal heartbeat; severe headache; blurry vision; fainting or loss of consciousness; weakness or sensory changes; confusion or trouble speaking; seizures; unexplained abdominal pain; or new leg pain or swelling” [14].

Acute myopericarditis associated with COVID-19 vaccine can be associated with arrhythmia. Anticipatory care includes judicious triaging of the disposition of patients seen in ambulatory or emergency setting for work-up or management of probable myopericarditis patients. Whenever possible, serology and PCR for common viral causes of myocarditis such as parvovirus B19, herpesvirus type-6, adenovirus, enterovirus, Epstein–Barr and cytomegalovirus should be obtained to rule out common causes of viral myocarditis. The treatment considerations for COVID-19 vaccine-associated myopericarditis include anti-inflammatory medications and guideline-directed medical therapy if left ventricular function is reduced [31]. No data for any specific treatment for vaccine-associated myocarditis are available. Steroids are used for their potent anti-inflammatory action in cases where the patient has continued symptoms and troponin leak even after NSAIDs. However, steroids and IVIG are also immunomodulatory and immunosuppressive agents and could reduce the specific immune response against SARS-CoV-2 triggered by the vaccine. Thus, the duration of steroids administration should be limited to the resolution of the symptoms or ventricular arrhythmias or the recovery of the LV function. Among 29 cases with a known outcome, all were discharged to their homes within 1 week, and all made full clinical recoveries. However, the long-term impact of myocardial inflammation following COVID-19 mRNA vaccine as detected on CMR, remains unknown and needs to be systematically evaluated. Further studies are required to elucidate the pathophysiology that underlies this complication to seek mitigation strategies and to delineate optimal therapy.

It is prudent to maintain regular follow-up of patients with COVID-19 vaccine-associated myocarditis patients especially those with documented inflammation on CMR. Pending publication of long-term outcome data after SARS-CoV-2 vaccine-related myocarditis, we suggest adherence to the current consensus recommendation to abstain from competitive sports for 3 to 6 months with re-evaluation before sports participation [32,33].

## 6. Conclusions

The preliminary data published suggest that all myopericarditis cases are mild and clinically resolve within a few days to a few weeks. The bottom line is that the risk of cardiac complications due to SARS-CoV-2 infection far exceeds the minimal and rare risks of vaccination-related transient myocardial or pericardial inflammation. The CDC accepts all reports through VAERS from everyone regardless of the plausibility of the vaccine causing the event or the clinical seriousness of the event. The VRBPAC will follow up on all reported cases to the VAERS and VSD and adjudicate the cases as per the CDC case definition of myocarditis and pericarditis. In the future, with continuous monitoring and evaluation of the reported data, we will have evidence-based information available to guide us for proper management and find a specific strategy to mitigate vaccine-associated myocarditis and pericarditis.

## Figures and Tables

**Figure 1 children-08-00607-f001:**
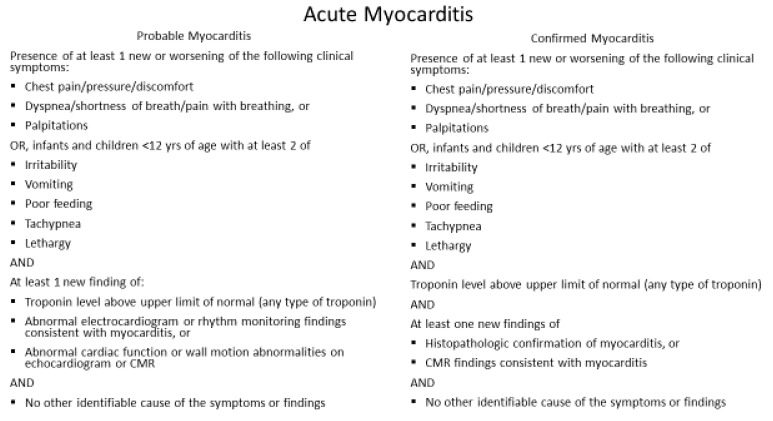
CDC working case definition for acute myocarditis. Modified from source: http://www.cdc.gov; accessed 18 June 2021.

**Figure 2 children-08-00607-f002:**
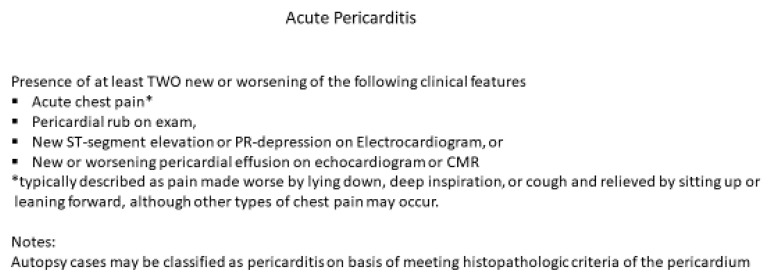
CDC working case definition for acute pericarditis. Modified from source: http://www.cdc.gov; accessed 18 June 2021. * Typically described as pain made worse by lying down, deep inspiration, or cough and relieved by sitting up or leaning forward, although other types of chest pain may occur.

**Table 1 children-08-00607-t001:** Summary of 29 patients (children and adults) of vaccine-related myocarditis and pericarditis published in the literature.

Age Years and Gender	mRNA Vaccine Dose	Days from Vaccination to Sx	COVID Serology	Symptoms	CRP	Tn	ECG	ECHO	CMR	Treatment	Hospitalization in Days
1. 16 M [9]	2nd	2	+	Fever Chest pain	↑	↑	ST↑ Junctional rhythm	NL LV Fx	LGE+	IVIG Steroid NSAID	6
2. 19 M [9]	2nd	3	-	Fever, Myalgia, Weakness	↑	↑	Diffuse ST changes	NL LV Fx	LGE +	NSAID Colchicine	2
3. 17 M [9]	2nd	2	+	Chest pain	↑	↑	Diffuse ST changes	NL LV Fx	Edema+ LGE+	NSAID	None
4. 18 M [9]	2nd	Immediately after vaccine	+	Chest pain	NA	↑	ST↑	NL LV Fx	NA	IVIG NSAID	3
5. 17 M [9]	2nd	3	+	Chest pain Fever	NA	↑	ST↑	NL LV Fx	NA	IVIG Steroid	3
6. 16 M [9]	2nd	3	+	Fever Chest pain	NA	↑	ST↑	NA	Edema LGE+	IVIG Prednisone	3
7. 14 M [9]	2nd	3	-	Fever Chest pain	NA	↑	Diffuse ST changes	LV Fx ↓	Edema LGE+	NSAID	3
8. 17 M [10]	2nd	3	NA	Chest pain	↑	↑	ST↑ RBBB	NL LV Fx	NA	NA	4
9. 16 M [10]	2nd	1	+	Chest pain	↑	↑	Diffuse ST changes	NL LV Fx PE	NA	NA	6
10. 16 M [10]	2nd	2	NA	Chest pain	↑	↑	Diffuse ST changes	LV Fx ↓	NA	NA	6
11. 16 M [10]	2nd	3	+	Chest pain Nausea	↑	↑	ST↑	NL LV Fx PE	NA	NA	4
12. 17 M [10]	2nd	1	+	Chest pain Headache	↑	↑	ST↓	NL LV Fx	NA	NA	5
13. 16 M [10]	1st	2	+	Chest pain Fever Dyspnea	↑	↑	ST↑	NL LV Fx	NA	NA	5
14. 17 M [10]	2nd	3	+	Chest pain Dyspnea	↑	↑	NL	NL LV Fx	NA	NA	3
15. 28 M [11]	Janssen	5	NA	Chest pain	NA	NA	ST↑	LV Fx ↓	LGE +	NA	NA
16. 39 M [9]	2nd	3	NA	Chest pain Dyspnea	NA	NA	Inverted T wave	RV Fx ↓	LGE+	NA	NA
17. 39 M [11]	2nd	4	NA	Fever Chest pain Dyspnea	NA	NA	NL	NL LV Fx	LGE+	NA	NA
18. 24 M [11]	1st	7	NA	Chest pain	NA	NA	NL	NL LV Fx	Edema+ LGE+	NA	NA
19. 19 M [11]	2nd	2	NA	Chest pain	NA	NA	Non-spec. ST changes	NL LV Fx	LGE+	NA	NA
20. 20 M [11]	2nd	3	NA	Chest pain	NA	NA	ST↑	LV Fx ↓	LGE+	NA	NA
21. 23 M [11]	2nd	3	NA	Fever Chest pain	NA	NA	Diffuse ST changes	NA	LGE+	NA	NA
22. 22 M [12]	2nd	3	NA	Fever Chest pain	↑	↑	Diffuse ST↑	LV Fx ↓	LGE+	NSAID Prednisone	NA
23. 31 M [12]	2nd	3	NA	Fever Chest pain Dyspnea	↑	↑	NL	LV Fx ↓	LGE+	No	NA
24. 40 M [12]	1st	2	NA	Chest pain	↑	↑	Diffuse ST changes	LV Fx ↓	LGE+	Prednisone Colchicine	NA
25. 56 M [12]	2nd	3	NA	Chest pain	↑	↑	Peak T waves	NL LV Fx	LGE+	Np	NA
26. 26 M [12]	2nd	3	NA	Fever, cough Chest pain	NA	↑	ST↑	LV Fx ↓	LGE+ PE	Colchicine	2
27. 35 M [12]	2nd	2	NA	Fever Chest pain	↑	↑	Diffuse ST changes	LV Fx ↓	Edema LGE+	NSAID	4
28. 21 M [12]	2nd	4	NA	Fever Chest pain	↑	↑	Diffuse ST changes	LV Fx ↓	Edema LGE+	NSAID	NA MCS NS VT
29. 22 M [12]	2nd	4	NA	Chest pain	↑	↑	Diffuse ST changes	LV Fx ↓	Edema LGE+	NA	NA NS VT

Sx: symptoms, Tn: troponin, ECG: electrocardiogram, RBBB: right bundle branch block, Non-spec.: non-specific, ECHO: echocardiography, CMR: cardiac magnetic resonance, M: male, NA: not available, NL: normal, LV: left ventricle, RV: right ventricle, Fx: function, LGE: late gadolinium enhancement, PE: pericardial effusion, NS VT: non-sustained VT, ↑= elevated, ↓= decreased.

## Data Availability

Data are publicly available as cited in the references.

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
