# Peer review of "Myocarditis and Pericarditis Following mRNA COVID-19 Vaccination: What Do We Know So Far?"

_children, 2021, doi:10.3390/children8070607_

Round 1
Reviewer 1 Report
Bibhuti et al present a review of the literature on Myocarditis and Pericarditis following mRNA COVID-19 vaccination. The subject is important in the context of expanding the vaccine target in children. A total of 29 cases were found in the literature. The clinical presentation and the biological parameters that should lead to this diagnosis are specified. The outcome was satisfactory for all cases; the benefit-risk balance is discussed. The review is synthetic and well written.
This review is worth publishing subject to minor changes
Introduction:
Line 32: Since the review is intended for an international readership, please do not limit the context to the United States of America. You can read the conclusions of the European Medicines Agency here: https://www.ema.europa.eu/en/news/comirnaty-spikevax-possible-link-very-rare-cases-myocarditis-pericarditis
Line 76: As of end of May 2021, cases of myocarditis reported in the EEA from the EudraVigilance database were: 38 for Vaxzevria and 0 for COVID-19 Vaccine Janssen. Cases of pericarditis were: 47 for Vaxzevria and 1 for COVID-19 Vaccine Janssen.
Moreover one does not understand why the subchapter 3 begins with this sentence
Line 85: the sentence is not finished
Line 104:
The European Medicines Agency has reported other cases: “This included an in-depth review of 145 cases of myocarditis in the European Economic Area (EEA) among people who received Comirnaty and 19 cases among people who received Spikevax. PRAC also reviewed reports of 138 cases of pericarditis following the use of Comirnaty and 19 cases following the use of Spikevax.”
If detailed information is not available and cannot be included in the review, these cases should at least be cited. Detailed information can be requested from the agency.
Line 103: it would be better to separate the presentation of myocarditis and pericarditis
Table1: the publication reference for each case could be added in the table
Line 109: what about forceful heartbeat reported elsewhere?
Tables 2 and 3 are not tables but screenshots. The image is unreadable. Please replace with a box, rewritten for the occasion
A methodology section is missing that explains how the cases were identified. Please submit the information required for a literature review.
Author Response
Thank you very much to the esteemed reviewer and the manuscript presntation has improved significantly with the expert comments and correction of errors. We have included the following response:
Introduction:
Line 32: Since the review is intended for an international readership, please do not limit the context to the United States of America. You can read the conclusions of the European Medicines Agency here: https://www.ema.europa.eu/en/news/comirnaty-spikevax-possible-link-very-rare-cases-myocarditis-pericarditis
- Thank you very much. We added the information from possible link of myocarditis and pericarditis with Comirnaty and Spikevax, “As of end of May 2021, cases of myocarditis reported in the EEA from the EudraVigilance database were: 38 for Vaxzevria and 0 for COVID-19 Vaccine Janssen. Cases of pericarditis were: 47 for Vaxzevria and 1 for COVID-19 Vaccine Janssen. The exposure in the EEA for each vaccine was around 40 million for Vaxzevria and 2 million for COVID-19 Vaccine Janssen.”
Line 76: As of end of May 2021, cases of myocarditis reported in the EEA from the EudraVigilance database were: 38 for Vaxzevria and 0 for COVID-19 Vaccine Janssen. Cases of pericarditis were: 47 for Vaxzevria and 1 for COVID-19 Vaccine Janssen.
- Thank you very much. See above.
Moreover one does not understand why the subchapter 3 begins with this sentence
- Thank you again, with your expert suggestion and as we added the above information, changed the first sentence.
Line 85: the sentence is not finished
- It was a typo error, dose of Moderna—should be continuous and is fixed.
Line 104:
The European Medicines Agency has reported other cases: “This included an in-depth review of 145 cases of myocarditis in the European Economic Area (EEA) among people who received Comirnaty and 19 cases among people who received Spikevax. PRAC also reviewed reports of 138 cases of pericarditis following the use of Comirnaty and 19 cases following the use of Spikevax.”
- Thank you very much, we added the information from EMA’s safety Committee (PRAC).
If detailed information is not available and cannot be included in the review, these cases should at least be cited. Detailed information can be requested from the agency.
- Thank you very much, I agree. We included information as above.
Line 103: it would be better to separate the presentation of myocarditis and pericarditis
- The presentation of myocarditis and pericarditis are sometimes difficult to separate: we used the term myopericarditis and separated wherever possible.
Table1: the publication reference for each case could be added in the table
- Thank you, we added the reference for each case in Table.
Line 109: what about forceful heartbeat reported elsewhere?
Tables 2 and 3 are not tables but screenshots. The image is unreadable. Please replace with a box, rewritten for the occasion
- Thank you, we redid theTable-2 and -3 using Power point and the revised Tables attached.
A methodology section is missing that explains how the cases were identified. Please submit the information required for a literature review.
- Thank you. We added a section in introduction and clarified how the cases were identified. The additional section is as follows: “We performed a systematic search on electronic databases (PubMed, Scopus, medRxiv and bioRxV) using the terms “mRNA vaccine complications” , “heart inflammation with COVID-19 vaccine”, “impact of COVID-19 vaccine in children and young adults”, “myocarditis after COVID-19 vaccination” and “pericarditis after COVID-19 vaccination”, and “myopericarditis after COVID-19 vaccination.”
Reviewer 2 Report
I read carefully the present paper concerning the reported cardiac side effects of mRNA COVID-19 vaccine.
It is an interesting and informative manuscript in a hot topic.
However, reading the manuscript i have the following questions.
Nine persons had no data from MRI (erroneously written in section 4, that 21 pts had MRI instead of twenty and it must be corrected) 7 pts haven't data from troponin levels making the diagnosis of myocarditis problematic!
Furthermore, there is a heterogeneity as concern the pts age with 6 of them be older than 30 years , 4 of them presented with reduced LV or RVEF (vs 8/23 in younger pts). This is no statistic but it's an observation.
The authors suggest that extremely high antibody levels of mRNA vaccines generate in young people, may inducing anti-idiotype cross-reactive antibody mediated cytokine expression in the myocardium causing myopericarditis. Has the authors any data of antibody levels in the reported cases?
I think that a study limitation section with these remarks must be introduced
Author Response
Thank you very much for your time and extremely useful comments. Our response as follows:
Nine persons had no data from MRI (erroneously written in section 4, that 21 pts had MRI instead of twenty and it must be corrected) 7 pts haven't data from troponin levels making the diagnosis of myocarditis problematic!
- Thank you. We corrected the numbers as picked-up by the reviewers. Regarding the 9 cases whose diagnosis is questioned, we cannot change as these are published data, but we add a comment
Furthermore, there is a heterogeneity as concern the pts age with 6 of them be older than 30 years , 4 of them presented with reduced LV or RVEF (vs 8/23 in younger pts). This is no statistic but it's an observation.
- Thank you. As we included all cases both children and adults, there is a lot of heterogeneity and we commented on this point.
The authors suggest that extremely high antibody levels of mRNA vaccines generate in young people, may inducing anti-idiotype cross-reactive antibody mediated cytokine expression in the myocardium causing myopericarditis. Has the authors any data of antibody levels in the reported cases?
- Thank you. This is an important point. As, these are published cases, we have no information on the antibody status. But, we included in the discussion that these should be obtained for diagnosis.
I think that a study limitation section with these remarks must be introduced
- Thank you very much. We added this also as limitation of the metanalysis.